# Molecular Beam Epitaxy of Layered Group III Metal Chalcogenides on GaAs(001) Substrates

**DOI:** 10.3390/ma13163447

**Published:** 2020-08-05

**Authors:** Sergey V. Sorokin, Pavel S. Avdienko, Irina V. Sedova, Demid A. Kirilenko, Valery Yu. Davydov, Oleg S. Komkov, Dmitrii D. Firsov, Sergey V. Ivanov

**Affiliations:** 1Ioffe Institute, 26 Politekhnicheskaya, 194021 St. Petersburg, Russia; avdienko.pavel@gmail.com (P.S.A.); irina@beam.ioffe.ru (I.V.S.); demid.kirilenko@mail.ioffe.ru (D.A.K.); valery.davydov@mail.ioffe.ru (V.Y.D.); 2St. Petersburg Electrotechnical University “LETI”, 5 Prof. Popova, 197376 St. Petersburg, Russia; okomkov@yahoo.com (O.S.K.); d.d.firsov@gmail.com (D.D.F.)

**Keywords:** III–metal chalcogenides, 2D materials, GaSe, InSe, GaTe, quantum wells, molecular beam epitaxy, GaAs(001) substrate, photoluminescence, Raman spectroscopy, transmission electron microscopy, X-ray diffraction

## Abstract

Development of molecular beam epitaxy (MBE) of two-dimensional (2D) layered materials is an inevitable step in realizing novel devices based on 2D materials and heterostructures. However, due to existence of numerous polytypes and occurrence of additional phases, the synthesis of 2D films remains a difficult task. This paper reports on MBE growth of GaSe, InSe, and GaTe layers and related heterostructures on GaAs(001) substrates by using a Se valve cracking cell and group III metal effusion cells. The sophisticated self-consistent analysis of X-ray diffraction, transmission electron microscopy, and Raman spectroscopy data was used to establish the correlation between growth conditions, formed polytypes and additional phases, surface morphology and crystalline structure of the III–VI 2D layers. The photoluminescence and Raman spectra of the grown films are discussed in detail to confirm or correct the structural findings. The requirement of a high growth temperature for the fabrication of optically active 2D layers was confirmed for all materials. However, this also facilitated the strong diffusion of group III metals in III–VI and III–VI/II–VI heterostructures. In particular, the strong In diffusion into the underlying ZnSe layers was observed in ZnSe/InSe/ZnSe quantum well structures, and the Ga diffusion into the top InSe layer grown at ~450 °C was confirmed by the Raman data in the InSe/GaSe heterostructures. The results on fabrication of the GaSe/GaTe quantum well structures are presented as well, although the choice of optimum growth temperatures to make them optically active is still a challenge.

## 1. Introduction

Layered two-dimensional (2D) group III metal chalcogenides (MCs) (GaSe, GaTe, InSe, and others) are still of strong interest for the development of novel high-performance semiconductor devices due to their unique electronic and optical properties [1,2,3]. In particular, these materials have attracted great attention for the fabrication of field-effect transistors [4], solar cells [5,6], high-efficient photodetectors, and gas sensing devices [7,8]. InSe has significant potential for using in thermoelectric applications [9]. In addition, GaSe and GaTe are the nonlinear optical crystals and can be used for terahertz (THz) wave generation [10,11].

One of the main reasons for intensive studies of 2D layered materials is that they exhibit different properties in a few-layer and bulk forms [12,13,14], providing an additional degree of freedom in the development of electronic and optical devices. To date, most of the 2D layered semiconductors and heterostructures based on them are either prepared by mechanical exfoliation from bulk crystals or grown by chemical (physical) vapor transport (CVT, PVT) or chemical vapor deposition (CVD) techniques [15]. Although the sophisticated multilayered structures consisting of various 2D materials bonded to each other via weak van der Waals forces can be produced using the scotch-tape method (“realizing the idea that 2D materials can be stacked as Lego blocks” [16]), this technology can hardly be used for the fabrication of commercial devices. At the same time, the use of epitaxial growth techniques, in particular molecular beam epitaxy (MBE), potentially allows fabrication of 2D thin films and heterostructures with abrupt interfaces and a uniform wafer-scale film thickness. Moreover, when the epitaxial growth proceeds via van der Waals forces (the so-called van der Waals growth mode), the 2D materials can be grown on highly-mismatched substrates, even when symmetries of the grown film and substrate differ [17], which is important for integration with conventional 3D semiconductors.

The abilities of MBE for the 2D device fabrication were recently demonstrated by Yuan et al. [18], who fabricated highly efficient photodetector arrays with external quantum efficiency up to 62% and photoresponse time of 22 μs using few-layer GaTe films grown by MBE on 3-inch Si wafer. The layer-by-layer growth mode has also been achieved in MBE of GaSe on mica and Si(001) substrates [19]. Wafer-scale GaTe_x_Se_1−x_ films, grown by MBE on freshly cleaved mica or cleaned silicon substrates, were also reported by Liu et al. [20]. Thus, the opportunity of realization of high-quality films and heterostructures composed of 2D group III MCs by MBE has already been proved. However, due to a number of polytypes and additional phases, the synthesis of defect-free films is still a problem. It is also complicated by the much lower substrate temperatures used in MBE as compared to other epitaxial techniques.

In this paper, we present the results on MBE growth of thin films of III–VI layered MCs (GaSe, InSe, and GaTe) and their heterostructures on conventional GaAs(001) substrates, as well as studies of their structural and optical properties. The use of a combination of conventional and two-dimensional semiconductors in a single heterostructure allows expanding the range of potential applications. Although the first publications on MBE growth of group III MCs date back to the early 1990s [21,22], there are many gaps in understanding growth mechanisms, and there is still no universal recipe for the growth of high-quality optically active structures based on these layered materials.

## 2. Materials and Methods

### 2.1. Materials Properties

The layered chalcogenides of the MX type, where X = (Se, Te) and M = (Ga, In), consist of vertically ordered layers which are bonded together by weak van der Waals forces, and each layer contains four covalently-bonded atomic sheets in the sequence X-M-M-X.

GaSe is one of the most studied 2D layered III–VI semiconductors, which has an indirect bandgap in the visible range with an energy of ~2 eV at T = 300 K, which is only 25 meV lower than the direct bandgap [1,23,24,25]. GaSe films can crystallize in different polytypes (β, ε, γ, and δ), which differ from each other by stacking sequence of the layers [26]. γ-GaSe has a rhombohedral crystal lattice with the R3m space group and the lattice parameters a = 3.75 Å and c = 23.92 Å, whereas β-, ε, and δ-GaSe have a hexagonal crystal lattice with P6_3_/mmc, $P\overset{¯}{6}m2$, and P6_3_mc space groups, respectively [26]. Both hexagonal β- and ε-GaSe polytypes have the same lattice parameters a = 3.750 Å and c = 15.94 Å [1]. The formation of one or another polytype depends on both the technique and growth conditions used for the GaSe film fabrication (see, e.g., [27]). In particular, GaSe layers grown by MBE are usually of ε- or γ-polytype [19,28,29,30]. However, some domains of β-polytype were also observed in GaSe films grown by MBE on MoS_2_ cleaved surfaces [31]. The thickness of one monolayer (or, more accurately, one tetralayer (TL)) GaSe is ~0.8 nm for all polytypes.

As with many other 2D materials, GaSe is unstable and oxidizes in ambient conditions, which leads to a gradual degradation of its optical properties [32,33,34]. The Raman spectroscopy analysis showed that GaSe decomposes into Ga_2_Se_3_ and amorphous Se during oxidation [32,34]. Moreover, during the prolonged storage in air, the GaSe surface also undergoes visible transformations [35]. The oxidation of GaSe is a rapid process with a monolayer reaching an oxidized state almost immediately after exposure to air [34]. To prevent the atmospheric oxidation, the surface passivation using 50-nm-thick Al_2_O_3_ layer was proposed [33]. The effective protection is even more important if the grown GaSe layer consists of a large number of nanometer-scale nanoplatelets.

InSe is a direct-gap semiconductor with E_g_ = 1.25–1.3 eV at room temperature in the bulk form; however, there is some spread in the bandgap values reported in the literature for various polytypes [1,36,37,38,39]. There are three polytypes of layered InSe (β, γ, and ε), which differ in the symmetry and stacking sequence of the layers [1,36]. The crystal space groups of the respective InSe polytypes are the same as in the case of GaSe. The lattice constant as well as the distance between two neighboring layers are nearly the same for all polytypes: a ≈ 4.0 Å [36,40,41,42], d ≈ 0.84 nm [1,36,39,43]. The ε- and γ-InSe are the most widely studied InSe polytypes with a hexagonal unit cell. These two hexagonal phases are hardly distinguishable, as they exhibit a similar composition of the unit cell and the same lattice parameters. The γ-polytype has a rhombohedral unit cell with the lattice parameter c = 24.96 Å, which is 1.5 times larger than that of β-InSe and γ-InSe.

In contrast to the most commonly studied 2D materials, the direct-gap semiconductor GaTe (E_g_ = 1.7 eV in a bulk form at T = 300 K [1]) has a monoclinic crystal structure in the space group C2/m, the symmetry of which is less than hexagonal lattice structure, with lattice parameters a = 17.32 Å, b = 4.05 Å, c = 10.54 Å and β = 104.4° [1]. Low-symmetry 2D materials show significant in-plane anisotropy in their electrical, optical, and thermal properties [44]. Two-thirds of the Ga-Ga bonds in GaTe are perpendicular to the layer and the rest are parallel to the layer. The thickness of each GaTe layer is also ~0.8 nm [1,18,45]. A metastable hexagonal phase for GaTe has also been reported with the lattice parameters a = 4.06 Å and c = 16.96 Å (space group P6_3_/mmc) [46].

Similar to GaSe, GaTe is also characterized by a very high oxidation rate in air, which results in a gradual degradation of its optical properties. Moreover, recently Fonseca et al. demonstrated that GaTe exposure to air leads to a significant restructuring of the conduction band due to the incorporation and chemisorption of oxygen to tellurium with subsequent transformation into a disordered GaTe–O_2_ phase [47]. To prevent the atmospheric oxidation of the GaTe, the surface passivation with Al_2_O_3_ is required [33]. In this study, we stored the GaTe samples in vacuum. This method also substantially delays the degradation of optical properties, but, of course, is not practical for real-life applications.

### 2.2. Thin-Films Synthesis

Thin films of 2D-layered MCs were grown on epi-ready GaAs(001) substrates at a substrate temperature of T_S_ = 400–530 °C using a double-chamber MBE setup (SemiTEq, St. Petersburg, Russia). Standard Ga, In, and Te effusion cells as well as a Se valve cracking cell (Veeco, Plainview, NY, USA) with the cracking zone temperature T_Se_(cr) = 500 °C were used as molecular beam sources. The samples were grown either directly on thermally cleaned GaAs substrate or on 200-nm-thick GaAs buffer layer grown in a separate III–V growth chamber and transferred to the III–VI chamber through vacuum. In the former case, the substrates were first heated to T_S_ ~ 600 °C without an As overpressure to remove the surface oxide layer, which was controlled by reflection high energy electron diffraction (RHEED) measurements, and then the substrate was cooled to the deposition temperature. In the latter case, after the transferring of GaAs buffer to the III–VI growth chamber the substrate temperature was gradually increased to the growth temperature with the rate ~20 °C/min. When the substrate temperature was stabilized at a given value, the growth was started as soon as possible to avoid the reaction between the substrate surface and residual Se. In the case when the initial deposition temperature was below 500 °C, we used also the main shutter for additional protection of GaAs(001) growth surface. The VI/III flux ratio was controlled by measuring the beam equivalent pressures (BEPs) of the respective elements at the substrate position by using a Bayard-Alpert ion gauge. The growth parameters of GaSe and GaTe epitaxial layers are outlined below. The details of MBE growth of specific III–VI MCs and heterostructures are given in the corresponding parts of the paper.

### 2.3. Characterization

The samples were characterized by cross-sectional transmission electron microscopy (TEM), scanning electron microscopy (SEM), X-ray diffraction (XRD), Raman spectroscopy, and photoluminescence (PL) spectroscopy techniques. The growth of all layers and heterostructures was monitored in situ by using RHEED.

The Raman measurements were performed both at room and liquid nitrogen temperatures using a T64000 (Horiba Jobin-Yvon, Lille, France) spectrometer equipped with a confocal microscope. The line at λ = 532 nm of Nd:YAG laser (Torus, Laser Quantum, Inc., Edinburg, UK) and the 514.5 and 488 nm lines of Argon laser (Spectra-Physics, Inc., Mountain View, CA, USA) were used as the excitation sources. The laser power on the samples was as low as ~25–80 µW with a spot size of ~1 µm in diameter. The room-temperature PL spectrum of the GaSe/GaAs(001) layer was also measured using the described experimental setup.

The low-temperature PL measurements of the GaTe/GaAs(001) layer as well as temperature-dependent PL measurements of the InSe/GaAs(001) layer were carried out using experimental setup based on a Vertex 80 Fourier-transform infrared (FTIR) spectrometer (Bruker, Germany) equipped with a Si photodiode detector and a CaF_2_ beam splitter as described in Ref. [48]. The samples were placed into a Janis CCS-150 closed-cycle helium cryostat with a LakeShore 325 temperature controller, and the luminescence was extracted via CaF_2_ window and lenses. A 405 nm violet diode laser SSP-DHS-405 was used as the excitation source.

Micro-PL spectra of the GaSe/GaAs(001) layers were measured within the 10–110 K temperature range in a He-flow ST-500-Attocube cryostat. The spectra were recorded using a cooled CCD camera; the laser line at λ = 405 nm of CUBE laser (Coherent) was used as the excitation source. The laser power density on the sample was below 1 W/cm^2^.

Time-resolved measurements of the luminescence in ZnSe/InSe/ZnSe structures were performed at T = 77 K. The samples were excited by 405 nm radiation of CUBE laser (Coherent) with pulses of 45 ps duration at a repetition rate of 1 MHz.

The cross-section transmission electron microscopy (TEM) measurements were carried out using Jeol JEM-2100F microscope (the accelerating voltage 200 kV; the pixel resolution 0.19 nm). The samples were prepared by the standard procedure, i.e., by mechanical polishing with subsequent thinning by etching with Ar^+^ ions (4 keV). D2 Phaser powder diffractometer (Bruker, Germany) was used for the XRD measurements. The SEM measurements were used to estimate the thickness of the layers and to study the surface morphology of the grown layers. Both plan-view and cross-section SEM images of the grown samples were obtained by using CamScan microscope.

## 3. Results and Discussion

### 3.1. MBE Growth of GaSe/GaAs(001)

The basic peculiarities of GaSe MBE growth on GaAs substrates were established in the 1990s [49,50,51,52,53]. In the last decade, a new surge of interest in GaSe (and, of course, in other 2D layered materials) has arisen because of the discovery of graphene in 2004 [54]. However, as already noted, no recipe for growing the high-quality optically active structures was developed. The GaAs(111)B substrate seems to be a natural candidate for the growth of GaSe layers because of a relatively low lattice-mismatch of ~6% [21]. Indeed, the MBE growth of GaSe on a hexagonally symmetric, three-dimensional GaAs(111)B substrate has been shown to be free of the critical thickness problem, which is of a great importance for potential device applications [21,49]. However, the growth of GaSe on GaAs(111) is hardly achieved at growth temperatures above 500 °C, since GaSe is re-evaporated from the growing surface due to a weak van der Waals force [49]. As a result, no PL emission from the GaSe films was detected due to its rather poor crystallinity.

The van der Waals growth mode was also realized in MBE growth of GaSe/GaAs(001) at low temperatures T_S_ ≈ 400 °C, with the ***c*** axis of the growing GaSe layer oriented along the normal to the substrate surface [49]. At higher growth temperatures (T_S_ > 500 °C), the nucleation of GaSe on the GaAs(001) surface occurs through the formation of transitional submonolayers with a chemical bonding between the substrate and the layer [49,50].

Both spiral and island growth modes were reported for GaSe layers grown by MBE on different substrates [31,52,55,56]. In particular, the spiral pyramid-like morphology has been reported for the GaSe/GaAs(001) layers grown by MBE at low T_S_ ≈ 400 °C [55]. The evolution of GaSe morphology with growth time, observed by using the atomic force microscopy (AFM) technique, has been explained by the realizing of screw-dislocation-driven (SDD) growth mode, which is typical for the growth of 2D materials [57]. We did not use AFM in this study systematically; however, our data on GaSe/GaAs(001) layers are in a good agreement with those of other research groups. Thus, the existence of SDD growth mechanism in our GaSe layers grown at T_S_ ≈ 400 °C directly on GaAs(001) substrates is highly likely. To grow uniform large-area layered materials with a controlled number of layers the SDD growth mode should definitely be avoided. According to Nie et al. [57], to do this, both the growth rate and the chalcogen-to-metal ratio need to be limited during deposition, and these parameters are especially important at the early nucleation and growth stages. The quality of the substrate also plays a significant role, so the use of a GaAs buffer layer along with a low growth rate as well as a choice of appropriate growth conditions could, in principle, provide the layer-by-layer growth mode in MBE of layered chalcogenides. The need for a low deposition rate for the formation of flat GaSe layers was confirmed by Yuan et al. [19], who demonstrated layer-by-layer growth of GaSe on mica and Si(001) substrates with a growth rate in the range 0.3–0.8 nm/min. The importance of a very low GaSe deposition rate (r_GaSe_ ≈ 0.2 nm/min) for the controllable growth of high-quality atomically thin GaSe layers was also demonstrated by Chen et al. [58].

The need for a low Se/Ga flux ratio is confirmed by the experimental fact that the optimum conditions for the MBE growth of GaSe on GaAs(001) substrates correspond to the nearly stoichiometric ratio between the adsorbed Ga and Se adatoms on the growth surface [30,49,50]. An analysis of the available literature data, taking into account the dependence of Se incorporation coefficient on both substrate and cracking zone temperatures of the Se valve source [59], shows that the stoichiometric conditions can be estimated as corresponding to P_Se_/P_Ga_ (BEP) ~ 12 and ~ 25 at T_S_ = 400 °C and 500 °C, respectively, if standard Ga and Se valve cracking cells with T_Se_(cr) = 500 °C are used as molecular beam sources [28]. In our analysis, we used the measured P_Se_/P_Ga_ (BEP) flux ratios needed to obtain a relatively smooth GaSe surface at various temperatures (350–450 °C) from [30], where a Se valve cracking cell with T_Se_(cr) = 950 °C was applied as a Se source, as well as data on GaSe layers grown by MBE using standard Ga and Se cells [29,49,50,51]. The estimated dependence of the R = P_Se_/P_Ga_ (BEP) ratio, corresponding to the stoichiometric conditions on the growth surface for various T_S_, is presented in Figure 1. One should note that it is difficult to associate the Se/Ga BEPs ratios taken from different sources with the stoichiometric growth conditions, especially keeping in mind that there is some uncertainty in determining the real temperature of epitaxial growth. However, it can be assumed that the formation of gallium droplets on the GaSe surface is the only reliable indicator of Ga-rich MBE growth conditions. The points corresponding to the GaSe layers grown at different T_S_ with and without Ga droplets on the growth surface are shown in Appendix A. The obtained data confirm the correctness of our estimations. It follows directly from the dependence in Figure 1 that the stoichiometry on the growth surface at a fixed T_S_ can be controlled either by adjusting the opening of the Se valve or by changing the cracking zone temperature T_Se_(cr) of the valve Se cell. In the latter case, the change in the ratio of the Se and Ga adatoms on the growth surface occurs due to a change in the Se sticking coefficient.

The growth of GaSe on GaAs(001) substrates was initiated by the simultaneous opening of Ga and Se fluxes. One should note, the RHEED observations showed that initial (2 × 4)As reconstruction of the GaAs(001) buffer layer changed to the (2 × 1)Se-terminated one while rising the substrate temperature under Se exposure or unintentionally if there was a Se background in the growth chamber. The growth parameters as well as the thicknesses of the GaSe epitaxial layers are outlined in Table 1. The presence or absence of a buffer layer in a particular structure is also indicated.

The surface morphology of the GaSe/GaAs(001) layers was analyzed by SEM. The plan-view images of the thick (>50 nm) layers grown at T_S_ ~ 400 °C and at different Se/Ga flux ratios with nearly the same growth rate of ~1.1–1.4 nm/min look quite similar: one can see a number of so-called “nanoplatelets” of different size and shape, randomly distributed on the relatively flat growth surface (Figure 2a,b). The similar surface morphology was observed in GaSe layers grown on GaAs(112) substrates at the same growth temperature T_S_ ~ 400 °C [28]. At the optimized conditions, where the adsorbed Ga and Se adatoms are close to stoichiometry, and the growth rate is relatively low, the RHEED pattern remained streaky for a long-time during MBE growth (Appendix A). However, under strong Se-rich conditions the brightness of the RHEED pattern gradually decreased with increasing layer thickness, and the streaky reflections became wider and blurred. In some areas of the RHEED pattern, spots as well as Debye rings appeared, indicating a polycrystalline growth. The observed RHEED images are very similar to that reported in [30] for the GaSe layers grown on GaN/sapphire in Se-rich conditions, and reflect an increase in surface roughness as the layer grows, which can be induced by the formation of a multi-domain structure at the initial growth stage. We assume that the “growth window” in the case of MBE growth of GaSe using standard Ga and Se valve cracking cells as molecular beam sources is narrower than in the case of using standard Ga and Se effusion cells.

Although the Se/Ga flux ratio affects both the surface density of nanoplatelets and their size, the main parameter determining the surface morphology, in addition to T_S_, is the Ga flux which controls the GaSe growth rate. The SEM image of the GaSe layer grown with a deposition rate of ~5 nm/min at T_S_ ~ 400 °C, as shown in Figure 2c, reveals a very rough surface of GaSe film despite the nearly stoichiometric flux ratio used. We suppose that the mobility of gallium atoms in this case is apparently insufficient for the formation of the flat layer morphology.

The cross-section TEM image of the GaSe layer grown at T_S_ ~ 400 °C with a low growth rate r_GaSe_ ≈ 1.13 nm/min (sample #*GS1*) confirms the formation of a relatively flat layer with a nearly abrupt GaSe/GaAs(001) interface (Figure 3a). The thickness of the GaSe film is about 200 nm and relatively uniform. The inhomogeneous contrast of the image can be related to the high density of extended defects and a number of domains in the grown film. Figure 3b shows the high-resolution TEM image of a GaSe/GaAs(001) interface for the same layer. The rough boundary between GaSe and GaAs is likely due to a thermal cleaning of the GaAs(001) substrate in the absence of the As flux. One can clearly see the layered structure of GaSe. The observed interplanar spacing of the film is of ~0.8 nm, which is equal to the thickness of one tetralayer. The selected area electron diffraction (SAED) (Appendix A) revealed that the layer has a predominantly rhombohedral structure, which corresponds to the γ-polytype. The formation of the γ-GaSe polytype is typical for the MBE grown GaSe/GaAs(001) layers, as reported previously [29,50]. The ***c*** axis of the GaSe layer is normal to the substrate/layer interface (Figure 3b). It is difficult to draw an unambiguous conclusion about the formation of any transition layer at the GaSe/GaAs(001) interface from the TEM image shown in Figure 3b. Nevertheless, the existence of built-in electric fields generated at the GaSe/GaAs(001) heterointerface, as indicated by clearly observed Franz–Keldysh oscillations in the photoreflectance spectra [60], may indirectly indicate the formation of such layers at the initial growth stages. Another possible explanation is related to the diffusion of Se atoms into the GaAs substrate. The possibility of the formation of Ga_2_Se_3_-type intermediate layers at the GaSe/GaAs(001) interface was demonstrated by Dai et al. [29,61].

With increasing T_S_, the surface morphology of the GaSe/GaAs(001) layers became rougher. It should be noted that this tendency was also observed for the GaSe layers grown in a two-stage regime, i.e., for layers in which the initial GaSe layer was deposited at a low substrate temperature T_S_ ~ 400 °C. At T_S_ ~ 500 °C, no flat layer was observed; the structure comprised an array of nanoplatelets tilted toward both $\left[1\overset{¯}{1}0\right]$ and $\left[\overset{¯}{1}10\right]$ directions of the GaAs substrate (Figure 4a and Appendix A) [49]. Despite a certain degree of disordering, due to the chemical interaction between the substrate and the growing layer, the ***c*** axis of GaSe nanoplatelets is predominantly oriented along the <111> directions of the GaAs substrate (Figure 4b). One can also see the formation of an intermediate layer near the GaSe/GaAs interface in some nanoplatelets, which leads to an additional layer tilt caused by a partial stress relaxation to accommodate the lattice mismatch between GaSe and GaAs, as previously reported by Kojima et al. [50]. Nevertheless, the areas where the GaSe domains start to grow with the ***c*** axis normal to the substrate/layer interface are retained (Appendix A). The average growth rate of the GaSe layer shown in Figure 4 was as high as ~2.2 nm/min. The appearance of the clearly resolved spots on the streaked RHEED pattern reflected the disordering on the growth surface. Nevertheless, the RHEED pattern remained nearly streaky during all 3 h of growth, and the brightness of the RHEED pattern remained virtually unchanged with increasing layer thickness (Appendix A). When the growth temperature was raised up to T_S_ > 530–540 °C, no growth of GaSe was observed even at a high Ga flux intensity.

For the GaSe layers grown at T_S_ ~ 400 °C with a Ga beam flux of (4–6) × 10^−8^ Torr, the growth rate estimated from the cross-section SEM images is nearly proportional to the impinging Ga flux, which reflects the growth of the layers with a relatively planar surface morphology (samples #*GS1*, #*GS2,* and #*GS6* in Table 1). However, for the layers with rough surface morphology grown at a high Ga beam flux, and for the layers consisting of a set of inclined nanoplatelets (*#GS7*), the average growth rate estimated from cross-section SEM images is higher than the expected one from the impinging Ga flux, which confirms the growth of discontinuous layers.

The X-ray powder diffraction patterns of the studied GaSe/GaAs(001) layers are presented in Figure 5. XRD patterns in addition to the reflections from the (002) and (004) planes of GaAs substrate demonstrate the reflections from GaSe planes oriented perpendicular to the ***c*** axis, which indicates the preferential orientation of the ***c*** axis in the direction normal to the growth surface. A weak additional peak in the angle range ~28.4° along the 2θ axis on XRD patterns of the layers grown under strong Se-rich conditions (Curves 1 and 2), can be attributed to the inclusions of a Ga_2_Se_3_ phase [30], since its position is in good agreement with the Ga_2_Se_3_(111) reflection [62], and the intensity is higher for the layer grown at the higher Se/Ga flux ratio. The preferred orientation of the Ga_2_Se_3_ phase is due to the proximity of interatomic distances in the GaSe(0001) and Ga_2_Se_3_(111) planes (i.e., along GaSe〈1000〉 and Ga_2_Se_3_〈110〉 directions), being 3.755 and 3.83 Å, respectively. According to Afifi et al. [62], the lattice constant of cubic α-Ga_2_Se_3_, determined from the analysis of the XRD patterns, is a = 5.44 Å. The existence of the Ga_2_Se_3_ phase inclusions in the GaSe layers grown under strong Se-rich conditions was also confirmed directly by TEM (Appendix A).

In contrast to that, XRD patterns of the samples grown at nearly stoichiometric conditions (Curves 3–5 in Figure 5) demonstrate the absence of the Ga_2_Se_3_ peak. The XRD patterns of the GaSe layers, which possess rough surface morphology (Curves 3 and 5 in Figure 5), contain additional peaks that can be interpreted as reflections from the inclined GaSe planes. The appearance of these additional peaks in the X-ray diffraction pattern probably evidences the internal ordering or, in other words, indicates the existence of preferred tilt angles of the nanoplatelets.

The GaSe layers were studied also by Raman spectroscopy techniques. Figure 6 shows the Raman spectrum of the bulk ε-GaSe sample produced by HQ Graphene (Groningen, the Netherlands) (the bottom curve), which was measured in a scattering configuration with the incident light directed along the ***c*** axis, as well as the typical Raman spectra of the GaSe/GaAs(001) epilayers grown at different conditions. The Raman spectra were measured under 532 nm laser excitation. It can be seen that, in the spectra of all samples, in addition to the lines corresponding to vibrational modes in GaSe [63], there are two extra lines at frequencies of 268 and 291 cm^−1^, which are associated with light scattering by transverse (TO) and longitudinal (LO) phonons in the GaAs substrate. However, in contrast to the spectrum of bulk ε-GaSe containing only one line near 213 cm^−1^, a distinct doublet is observed in this spectral region in the spectra of the GaSe layers grown at T_S_ ~ 500 °C (Curves 4 and 5), as well as in the spectrum of the GaSe layer with the rough surface morphology grown at T_S_ ~ 400 °C (Curve 3, #*GS8*). According to the selection rules [63,64], the line in the doublet, which we interpret as Eʹʹ mode, should appear in the spectrum, provided that the incident light is not directed strictly along the ***c*** axis. Thus, the presence of such a doublet is an indication of the substantial deviation of the ***c*** axis in the GaSe nanoplatelets from the normal to the substrate/layer interface. In addition, a broad peak with a frequency of about 250 cm^−1^ is observed in the spectra of the same samples. There are several explanations for its origin. This peak can be an Eʹ(LO) mode from ε-GaSe, which manifests itself in the spectra as a result of the deviation of the ***c*** axis in the GaSe nanoplatelets from the normal to the substrate/layer interface, or it can arise due to symmetry breaking caused by imperfection of the crystal lattice [65,66]. On the other hand, this peak can also be associated with amorphous Se, which can exist on the surface, since GaSe is easily oxidized [32,67]. Thus, the obtained Raman data presented by Curves 3–5 in Figure 6 allow us to conclude that these layers predominantly possess an ε-GaSe polytype. The SAED measurements of the sample #*GS3* agree well with this assumption, indicating the hexagonal crystal structure of the grown layer. An additional argument in favor of this interpretation is the first-principle calculations which predict the ε-polytype being a more stable form in comparison with that of β-polytype also having a hexagonal crystal structure [65,68].

Curve 2 in Figure 6 shows the typical Raman spectra of flat GaSe layers grown at T_S_ ~ 400 °C under strong Se-rich conditions (sample #*GS1*). One can see that the fundamental difference between this spectrum and the spectrum of bulk ε-GaSe is the absence of a line at a frequency of 19 cm^−1^. According to the group-theory analysis, the lowest frequency line in the Raman spectrum of GaSe should correspond to interlayer vibrations, and therefore should not be observed in the first-order Raman spectra of γ-GaSe because the primitive unit cell of γ-GaSe contains only one layer per unit cell [69]. Another peculiarity of Curve 2 in Figure 6 is that two lines in the spectrum are shifted to low frequencies compared to similar lines in the spectrum of bulk ε-GaSe, which are located at frequencies of 134 and 214 cm^−1^ (see also Appendix A). A similar shift of the lines was reported also by Yuan et al. [19], where the correspondence of the grown layer to the γ-GaSe polytype was determined using TEM. An analysis of the Raman data for both ε- and γ-polytypes of GaSe presented by Hoff et al. [64] also leads to the conclusion that the low-frequency shift of the two phonon modes is a characteristic of the γ-GaSe polytype. Thus, the peculiarities in the Raman spectrum of the GaSe layer grown at T_S_ ~ 400 °C (Curve 2 in Figure 6) confirm the conclusion drawn earlier from SAED (Appendix A) that sample #*GS1* is the γ-GaSe polytype. Summarizing, from the analysis of the Raman spectra of GaSe samples, we can conclude that the formation of a particular GaSe-polytype is determined not only by the epitaxial growth temperature, but rather by a combination of several factors.

Figure 7a demonstrates the temperature-dependent PL spectra of the γ-polytype GaSe layer grown at T_S_ = 410 °C in Se-rich regime (sample #GS1). As can be seen in the figure, PL spectra at different temperatures demonstrate broad emission bands centered at around 1.7 eV. The peak near 1.5 eV is attributed to the emission from GaAs substrate. The main peak at 1.7 eV apparently consists of at least two peaks and its intensity rapidly quenches with increasing temperature. The observed PL spectra are very similar to those reported by Diep et al. for the GaSe/GaAs(001) layers also grown at T_S_ = 400 °C [55], where the PL peaks were attributed to the free and bound excitons, respectively, and the strong redshift in PL was explained by the strain-induced reduction of the GaSe bandgap energy due to the SDD growth mode. The MBE growth conditions of the GaSe layers in [55] are similar to those used for growing sample *#GS1* (see Table 1), where the occurrence of SDD growth mode is definitely highly likely due to the high Se/Ga ratio. However, when tensile strain is applied a noticeable shift of both A_1_ compressional modes corresponding to the stretching vibrations of the atoms and located at frequencies of 134 cm^−1^ and 307 cm^−1^ should be observed [70,71]. Nevertheless, one can see no significant shift of the 307 cm^−1^ line in comparison with that of the bulk sample (Curves 1 and 2 in Figure 6). In addition, the observed rapid decay of the PL intensity with temperature allowed assuming the defect-related nature of this PL band. This assumption is supported by the fact that an intense emission band centered at 1.71 eV (77 K) was observed by Capozzi and Montagna [72] in the PL spectra of “poor-quality” GaSe samples grown from the melt by using the Bridgman–Stockbarger method. For each temperature studied, this emission band was more intense and broader in the samples containing a higher density of structural defects or impurities with respect to the samples of good quality. Capozzi and Montagna proposed that this band originates from the recombination of free electrons, mainly at the indirect minimum, with neutral-acceptor levels [72]. A similar PL emission band was also observed in the PL spectra of GaSe/GaAs(001) layers grown at high T_S_ > 500 °C [49,73]. To make a reliable conclusion about the origin of the 1.7 eV peak in the PL spectrum of our GaSe layer #GS1, time-resolved PL measurements should definitely be carried out.

In contrast to that, the GaSe layer grown at high T_S_ ~ 500 °C (sample *#GS7*) demonstrates strong PL at T = 300 K (Figure 7b). The spectral position (1.99 eV) of the emission line corresponds well to the emission line of the direct free exciton at T = 300 K [74]. The strong anisotropic dependence of the PL intensity (the details can be found in [75]) is evidence of a strictly oriented array of GaSe nanoplatelets in this structure, which is in a good agreement with TEM data.

### 3.2. MBE Growth of InSe-Based Heterostructures on GaAs(001) Substrates

The main problem in the MBE growth of InSe layers is associated with the formation of numerous In_x_Se_y_-type phases [36,76]. The relation between the film components and Se/In flux ratio corresponds to that appearing in the phase diagram [36]. In other words, one can expect the coexistence of In_4_Se_3_ with the main InSe phase at low Se/In flux ratios, as well as the occurrence of In droplets on the growth surface, while at high Se/In flux ratios the In_2_Se_3_ phase will predominantly crystallize. Indeed, the existence of “parasitic” In_x_Se_y_ phases (mainly In_4_Se_3_) has been reported in the InSe layers grown in a wide range of growth conditions on different substrates: glass, Si(001), Si(111), GaAs(111), and GaAs(001) [22,73,77,78,79,80]. The most comprehensive study of the film composition as a function of both Se/In flux ratio and T_S_ was performed by Emery et al. for the InSe films grown by MBE on amorphous substrates using standard elemental In and Se effusion cells [22,79]. Chatillon et al. explained the experimental results by performing a thorough thermodynamic analysis of MBE growth in the In-Se system [76,81]. The difference between calculated and experimental impinging molecular fluxes was taken into account by introducing the Se incorporation coefficient, which is the usual way for the thermodynamic description of the MBE growth of Se-based semiconductors [82,83]. According to Brahim-Otsmane et al. [79], at the substrate temperature of T_S_ ~ 350 °C for Se/In BEP ratio R < 2, which corresponds to the nearly stoichiometric ratio between the adsorbed In and Se adatoms on the growth surface [81], mixed films consisting of InSe and In_4_Se_3_ phases are grown, while at R > 3 the grown film is composed of InSe and γ-In_2_Se_3_ phases. Moreover, the relatively narrow “MBE growth window” for InSe films (R = 2–3) becomes narrower with increasing growth temperature up to T_S_ ~ 450 °C, and the In_4_Se_3_ phase inclusions are observed even in InSe layers grown at nearly stoichiometric conditions [79,84]. The formation of In_4_Se_3_ phase was also observed during the growth of InSe layers on GaAs(001) substrates, as confirmed by Raman measurements [75].

The growing of InSe films by using a Se valve cracking cell is even more complicated task. It should be noted that Ga_2_Se_3_ phase inclusions were observed in GaSe/GaAs(001) layers grown at a high cracking zone temperature (T_Se_(cr) = 950 °C) of the Se valve cell even at nearly stoichiometric conditions [30]. In this case, the growth window for “pure” InSe becomes even narrower or, possibly, disappears completely, especially at high T_S_. The Se/In BEP ratio corresponding to the nearly stoichiometric conditions is also changed by virtue of a change in the Se sticking coefficient.

In our experiments on MBE growth of InSe/GaAs(001) layers, the standard In cell and Se valve cracking cell with a cracking zone temperature of T_Se_(cr) = 500 °C were used as molecular beam sources. The In beam flux was ~1.3 × 10^−7^ Torr (BEP), which corresponds to an average InSe growth rate of 1.2–1.3 nm/min. The layers were grown at T_S_ = 450 °C. This temperature was chosen to provide the maximum structural quality of growing film and simultaneously avoid the film re-evaporation. Suggesting the similar dependence of stoichiometric VI/III flux ratio on Se cracking zone temperature as in the case of GaSe, the nearly stoichiometric InSe/GaAs(001) MBE growth conditions at T_S_ = 450 °C were estimated as Se/In(BEP) ~5–6 (T_Se_(cr) ~ 500 °C). The InSe growth on GaAs(001) was initiated by the simultaneous opening of In and Se shutters.

The plan-view and cross-section SEM images of the InSe film grown directly on GaAs(001) substrate at T_S_ = 450 °C at nearly stoichiometric Se/In flux ratio are presented in Figure 8. One can see the film (hereinafter, sample #*IS1*) consists of a number of randomly oriented nanocrystallites, which confirms the difficulties in InSe nucleation on the GaAs(001) surface at high temperatures [73]. The different shape of the nanocrystallites may also indicate the presence of secondary phase inclusions.

The X-ray powder diffraction pattern of the InSe/GaAs(001) layer (sample #*IS1*) is presented in Figure 9. The most intense diffraction peaks are attributed to InSe (00l), which indicates that there are a lot of InSe crystallites whose **c** axis is perpendicular to the substrate plane. The significant part of additional peaks can be attributed as belonging to the In_4_Se_3_ phase [84,85], and the intensities of these peaks (marked by black circles) are comparable with that from InSe. We assume that the presence of a significant amount of the In_4_Se_3_ phase in the film is associated primarily with the MBE growth conditions used, but also with the use a valve cracking cell as a Se source. Due to the similar positions of (003n) and (002n) reflections in the XRD pattern of rhombohedral and hexagonal InSe crystals, it is difficult to conclude which InSe polytype occurs. The intense peak at 2θ = 29.00 can be an indication of rhombohedral symmetry, since reflection at this Bragg angle exists only in powder XRD of the rhombohedral InSe modification [86]. However, this peak can also be associated with (040) reflection of In_4_Se_3_. The existence of additional asymmetric reflections on XRD pattern agrees well with observed surface morphology of the film.

The InSe reflections in Figure 9 are indexed for the γ-InSe rhombohedral lattice, since in this case the positions of asymmetric peaks are found to fit better the measured XRD pattern; we used the lattice parameters of rhombohedral InSe reported by Nagpal and Ali [42]: a = 4.0046 Å and c = 24.960 Å.

Raman spectra of the InSe/GaAs(001) layer (sample #*IS1*) measured at 80 and 300 K are shown in Figure 10. The line at 532 nm of Nd:YAG laser was used as the excitation source. Five peaks located at 18, 41, 115, 177, and 227 cm^−1^ are clearly seen in the room temperature Raman spectrum. These values are typical for InSe phonon modes measured under off-resonance conditions [87]. The phonon frequencies increase by a few cm^−1^ with decreasing temperature from 300 to 80 K. Additional peaks in the low temperature Raman spectrum detected at 38, 72, 104, and 168 cm^−1^ can be associated with the presence of In_4_Se_3_ phase [73,84], which is in a good agreement with XRD measurements. As mentioned above, according to XRD data, the InSe/GaAs(001) layer has a rhombohedral lattice, which indicates a γ-InSe polytype. However, the presence in the Raman spectrum of a low-frequency line at 18 cm^−1^ is in contradiction with this statement. According to the group-theory analysis, the lowest frequency line in the Raman scattering spectrum of InSe should correspond to interlayer vibrations [88]. The primitive unit cell of γ-InSe contains only one layer per unit cell and therefore this line should not be observed in the first-order Raman spectra. The appearance of this line is an important signature of presence of either ε- or β-phase InSe [87]. On the other hand, in the Raman spectra of non-centrosymmetric γ- and ε-InSe polytypes, polar longitudinal (LO) modes should be observed in certain scattering geometries and the intensity of those modes is enhanced in resonance conditions [89,90,91].

To clarify the reason for the contradiction between the XRD and Raman data, the measurements were made in the near-resonance Raman scattering using 514.5 and 488 nm lines of Argon laser as excitation sources. For γ-InSe, the 514.5 nm line has energy of about 10 meV below the $E_{1}^{\prime }$ exciton energy, while the energy of the 488 line is 9 meV above the exciton energy at the respective temperatures [91]. The corresponding spectra are shown in Figure 11 together with the off-resonance Raman spectrum obtained using the line at 532 nm as an excitation source. It can be seen that at room temperature a new doublet feature with frequencies of 199 cm^−1^ and 211 cm^−1^ is observed upon excitation at 514.5 nm. In the spectrum at T = 80 K obtained upon 488 nm excitation, the doublet feature experiences a high-frequency shift and a significant increase in intensity. Similar spectra for γ-rich bulk InSe were reported for the first time by Kuroda and Nishina [91]. It should be noted that in this work, upon excitation by the 514.5 and 488 nm lines, a low-frequency line was detected in the region of about 17–18 cm^−1^ in the Raman spectra (similar to that shown in Figure 11). However, this line, as already discussed above, should not be observed for the pure γ-InSe.

The spectra shown in Figure 11 make it possible to exclude the presence of β-phase in the InSe/GaAs(001) layer, since polar (LO) phonon modes that could be resonantly enhanced are absent in the Raman spectrum of β-InSe polytype [90]. Thus, taking into account the XRD data, the InSe/GaAs(001) layer (sample #*IS1*) can be characterized as γ-rich InSe, implying that along with the main γ-phase of InSe there exists also a certain amount of ε-phase. This result is in good agreement with the fact that MBE grown InSe films are usually of γ-polytype [73,78,80,92,93].

The PL spectra of the InSe/GaAs(001) layer (sample #*IS1*) measured within the temperature range of 11–110 K are presented in Figure 12. A 405 nm violet diode laser SSP-DHS-405 was used as the excitation source; the excitation power was as low as 50 mW, which corresponds to the excitation power density of ~0.5 W/cm^2^. The highest-energy peak observed in the PL spectrum at T = 11 K is detected at ~1.307 eV. This is more than 30 meV lower than the exciton recombination line (1.338 eV) recently reported by Shubina et al. [94] for flakes freshly cleaved from high structural quality bulk InSe grown by the Bridgman–Stockbarger method. According to the previous results [95,96], in the region below 1.32 eV, three main broad bands around 1.31, 1.28, and 1.23 eV have been reported in undoped InSe, which were assigned to impurity-band, donor–acceptor pair, and impurity-vacancy complex transitions, respectively. In a good agreement with these reports, the low-temperature spectrum shown in Figure 12 (at T = 11 K) can be also approximated by at least three different Gaussian peaks centered at 1.307, 1.28, and 1.23 eV, respectively. The broad peak at 1.307 eV (T = 11 K), which dominates the spectrum at low temperatures, quenches quickly with increasing temperature and disappears at T ≈ 50 K. At temperatures higher than 40 K, the defect-related band begins to dominate in the spectra. However, at temperatures higher than 60 K, one can see an occurrence of a clearly resolved peak centered at ~1.314 eV. From the PL temperature dependence, we can assume that the initial broad peak centered at ~1.307 eV (T = 11 K) actually consists of two closely spaced peaks located at 1.301 and 1.315 eV. Indeed, the spectrum at T = 11 K is most accurately approximated by four Gaussian contours (Appendix A). Homs and Marí [97] performed a comparative study of the PL of undoped and slightly neutron-transmutation doped InSe samples. In the region around 1.31 eV in doped samples, they found two closely-spaced and poorly separated bands, which they indexed as *E* and *F*, respectively. These bands correspond well to *P1* and *P2* peaks in Appendix A. They found also that emission *F* thermally quenches faster than *E*. The peak *E* was attributed to the recombination of a native acceptor, while peak *F* was assigned to donor–acceptor pair transition where both donors and acceptors are ionized. Indeed, in our case, we observe a similar temperature dependence of *P1* and *P2* peaks (see Figure 12a) and, accordingly, we can accept the proposed interpretation. In the same way, the band at 1.28 eV also can be attributed to a donor–acceptor pair transition [96,97] due to its fast temperature quenching. Thus, these three transitions (*P1*, *P2*, and *P3*) could be assigned to InSe native impurities (defects) because they have been observed in undoped samples [97]. Finally, the broad band observed at ~1.23 eV at T = 11 K can be associated with structural defects in the crystal. The presence of this band is an indication of a high concentration of defects and this band is assigned to a transition within an impurity vacancy-complex [95,96].

To improve the surface morphology of the layer, it is apparently possible to use the two-stage growth technique, which involves the deposition of the InSe buffer layer at low T_S_ in order to form a more planar interface [98,99]. A more advanced strategy involves the use of a GaSe buffer layer [73,80]. To control the film orientation, Budiman et al. also used the GaAs(001) substrates misoriented by 2° or 5° toward [110] direction [73]. However, despite the relatively planar surface morphology of the GaSe buffer layer (see Section 3.1), the top InSe layer grown at high T_S_ ~ 450 °C is again composed of a number of microcrystallites (see Appendix A). The observation of Debye rings in the RHEED pattern during MBE growth of InSe also confirms the polycrystalline nature of the film.

The strong diffusion of Ga into the top InSe layer with the formation of the InGaSe alloy was observed in the InSe/GaSe/GaAs(001) heterostructure, where the InSe growth temperature was as high as T_S_ ≈ 465 °C. The RT Raman spectrum of this structure is presented in Figure 13 (upper curve) along with the Raman spectra of reference InSe/GaAs(001) (middle curve) and γ-GaSe/GaAs(001) (lower curve) layers. The Ga diffusion into the top InSe layer is confirmed by the shift of all InSe Raman peaks in comparison with the reference InSe/GaAs(001) layer. The weak peaks at 59 and 132 cm^−1^ in the upper curve in Figure 13 correspond to the underlying GaSe buffer layer. It should be noted that strong interdiffusion at the InSe/GaSe interface at the growth temperatures of InSe higher than 400 °C was observed also in previous works [73,80].

Taking account of difficulties in InSe/GaAs(001) heteroepitaxy, as well as the problems with high-temperature InSe growth on GaSe surface, we tried to realize the ZnSe/InSe/ZnSe quantum well (QW) heterostructures. The InSe/ZnSe looks as a promising heteropair for potential applications in the field of photonics and electronics, when taking into account the fact that the band line-up for the InSe/ZnSe interface was reported to be of strong type I [100] as well as the existence of matured MBE technology of ZnSe/GaAs(001) [59].

The nominal thickness of InSe layer in the ZnSe/InSe/ZnSe QW heterostructures ranged 2–7 TLs. The thicknesses of the ZnSe bottom and top layers were 80–160 and 20–40 nm, respectively. The growth conditions of InSe on ZnSe(001) surface were chosen the same as in the case of InSe/GaSe heteroepitaxy: T_S_ = 450 °C and flux ratio Se/In (BEP) ~ 5–6. The top ZnSe layer on the InSe surface was grown at T_S_ = 300 °C. The details of MBE growth of ZnSe/InSe/ZnSe QW heterostructures as well as studies of their surface morphology were recently reported by Avdienko et al. [101]. Here, we focus mainly on the structural and optical properties of these QW structures.

Figure 14a shows the cross-sectional high-resolution TEM image of the ZnSe/InSe/ZnSe QW structure with a 7-TL-thick InSe layer. The TEM data confirm the multi-domain character of InSe/ZnSe(001) nucleation. At the growth conditions used, the ***c*** axis of InSe is directed normal to the growth surface. The TEM data also confirm the (111)-oriented ZnSe growth on the InSe(0001) surface, as previously reported by Smathers et al. [102]. However, the orientation of the ZnSe layer remains unchanged in those parts of the structure where there are no InSe islands. The existence of the (111)-oriented ZnSe phase in the structure was also confirmed by SAED measurements [101]. We suppose that the growth of ZnSe with a (111) orientation onto the InSe(0001) van der Waals surface is caused by the proximity of the InSe and ZnSe(111) lattice parameters (Δa/a ~ 1% or less, a(InSe) = 4.005 Å [41]). Therefore, the InSe islands provide the epitaxial orientation of the upper ZnSe layer, despite the weak interaction between them. In this case, the island-like growth of ZnSe(111) can be explained in terms of minimizing the elastic energy of the growing layer [101].

All QW structures demonstrate similar PL independent from InSe nominal thickness. One can see the intense broad line centered at ~2.27 eV (Figure 14b), indicating the same origin of this line in all QW structures. The time-resolved PL measurements demonstrate that the PL decay time is large and also nearly the same (~270 ns) in all structures (Figure 14c). Therefore, we can assume the luminescence is associated with radiative recombination centers of In in ZnSe because of the In diffusion into the underlying ZnSe layer. This assumption agrees well with the experimental value of the acceptor level energy in ZnSe crystals doped with indium, which is ~0.41 eV above the valence band [103]. A similar PL line at ~2.25 eV was also previously observed in In-doped ZnSe crystals [104].

### 3.3. MBE Growth of GaTe Layers and GaTe/GaSe Heterostructures on GaAs(001) Substrates

As mentioned above, one of the main problem in the MBE growth of different layered III-VI’s is the formation of numerous “parasitic” phases, i.e., In_4_Se_3_ and In_2_Se_3_ [73,101] or Ga_2_Se_3_ [28,30] in the case of MBE growth of InSe and GaSe, respectively. For the Ga-Te system, the other possible stable phase is Ga_2_Te_3_ [105]. The Ga_2_Te_3_ films were successfully synthesized by metalorganic MBE on both InP(001) and GaAs(001) substrates at T_S_ = 400–500 °C under strong Te-rich conditions (Te/Ga ~ 30) [106,107]. However, no evidence of Ga_2_Te_3_ inclusions has been obtained in MBE grown GaTe/GaAs(001) epilayers at much lower (Te/Ga ~ 10) flux ratio [108].

In this study, GaTe layers were grown on GaAs(001) substrates via a 200-nm-thick GaAs buffer layer under Te-rich conditions at low T_S_ ≈ 450 °C (sample #*GT1*) and high T_S_ ≈ 530 °C (sample #*GT2*) substrate temperatures, respectively. The GaTe growth was initiated by the simultaneous opening of both Te and Ga fluxes onto the GaAs(001) surface. The Te/Ga flux ratios were chosen to provide the Te-rich conditions in accordance with Bae et al. [108]. The growth parameters as well as thicknesses of GaSe layers are outlined in Table 2.

As mentioned above, GaTe preferentially crystallizes in the monoclinic α-structure (m-GaTe), which is more stable than the hexagonal close-packed β-structure (h-GaTe) [108,109,110]. The metastable phase h-GaTe generally exists in ultrathin samples [109,111]. However, under appropriate growth conditions, thick hexagonal GaTe flakes can be also obtained. Yu et al. recently demonstrated the phase-controlled synthesis of monoclinic and hexagonal GaTe using conventional physical vapor deposition technique [112]. They found that growth temperature is the key factor for phase control of GaTe flakes, and h-GaTe flakes are intended to be stable under relatively low deposition temperature. The importance of low deposition temperature for h-GaTe fabrication is also in a good agreement with the growth conditions reported for the GaTe films grown by metal-organic chemical vapor deposition [113]. According to reported results, h-GaTe is a thermodynamically unstable phase and converts into monoclinic one (m-GaTe) with the increase in sample thickness or environmental temperature [108,109,113]. In particular, the thickness-dependent transformation of h-GaTe to m-GaTe has been reported for the GaTe layers grown by MBE on GaAs(001) substrates [108]. At substrate temperature T_S_ ≈ 450 °C and Te/Ga(BEP) = 10, the critical thickness of such transformation has been estimated as high as ~90 nm.

The growth temperature is the main factor affecting the properties of the III–VI layers. Generally, the layers grown at higher T_S_ demonstrate the better crystal and optical quality. The plan-view SEM images of the samples #*GT1* and #*GT2* grown at different T_S_ are presented in Figure 15. The surface morphology of sample #*GT1* (Figure 15a) is typical for the MBE grown III–VI layers and is very similar to the surface morphology observed in the GaSe layers grown at low T_S_ ~ 400 °C (see Figure 2). In contrast, the plan-view SEM image of sample #*GT2* (Figure 15b) exhibits pronounced surface relief anisotropy, which is either related to the strong anisotropic nature of the m-GaTe crystal structure or induced by the large lattice mismatch between the GaTe and GaAs(001) substrate.

An increase in the growth temperature can lead to chemical interaction between the substrate and the growing layer. This interaction in case of GaSe/GaAs(001) MBE growth (T_S_ > 500 °C) results in the significant surface morphology transformation because of the growth of inclined nanoplatelets (see Figure 4). However, the cross-section SEM image of GaTe layer (sample #*GT2*) (see inset in the Figure 15b) grown at high T_S_ ≈ 530 °C demonstrates the formation of a relatively flat layer with nearly abrupt GaTe/GaAs(001) interface.

The RHEED images during the growth of samples #*GT1* and #*GT2* are presented in Figure 16. One can see that the RHEED patters remain streaky for both samples. However, the reflections for sample #*GT1* grown at low temperature (T_S_ ≈ 450 °C) are thickened and blurred (Figure 16a,b), and the brightness of the RHEED pattern gradually decreased with increasing the layer thickness. This behavior reflects an increase in the surface roughness as the layer grows, and can be induced by the formation of multi-domain structure at the initial growth stage, i.e., the MBE growth of GaTe is similar to that observed for GaSe/GaAs(001) at low substrate temperatures. In contrast, the RHEED pattern of sample #*GT2* (T_S_ ≈ 530 °C) demonstrates much narrower streaky reflections (Figure 16c), confirming that the higher is the growth temperature, the higher is the crystal quality of the growing film.

To understand the difference between samples #*GT1* and #*GT2,* we performed PL measurements. The as-grown samples were transferred through the air from MBE load-lock chamber into a closed-cycle helium cryostat. After that, they were stored in vacuum at room temperature for three weeks. The PL spectra were measured at T = 11 K in a closed-cycle helium cryostat using a laser source with the excitation wavelength of λ_exc_ = 405 nm. The laser power on the sample was as low as 50 mW with a spot of ~0.1 cm^2^, which corresponds to the excitation power density of 0.5 W/cm^2^. PL spectra of sample #*GT2*, measured after different storage times (within three weeks) are shown in Figure 17. The spectra are almost identical, confirming the absence of degradation of the GaTe optical properties during vacuum storage.

The shortest wavelength peak located at ~1.72 eV could be associated with the emission of acceptor-bound excitons [114], while the broad band centered at 1.57 eV is presumably associated with donor–acceptor pair (DAP) recombination [115]. This interpretation is in good agreement with the assumption that Te-rich growth conditions should cause the GaTe layer to exhibit p-type natural conductivity [108]. One of the possible explanations of 1.45 eV peak is that it originates from the GaAs substrate. However, this peak was not observed in the PL spectrum of sample #*GT1*, and the PL maximum is shifted by ~50 meV toward a longer wavelength compared to the peak from the GaAs substrate. It is also highly unlikely that this peak is due to radiation from the hexagonal h-GaTe. Although the weak broad PL emission at 1.44 eV (T = 300 K) was observed by Fonseca et al. [116] from the hexagonal GaSe_x_Te_1−x_ islands with low Se content (x ≤ 0.05), according to Cai et al. [117], the hexagonal GaTe should possess the bandgap at 1.03 eV, which does not correlate with intense emission at 1.45 eV. Moreover, the X-ray powder diffraction pattern of sample #*GT2* (Figure 18a) indicates the dominance of monoclinic GaTe [108,110]. Cai et al. [118] argued that this peak is associated with the emission centers localized at the domain edges. The dominance of this emission band (1.45 eV) in our PL spectrum, measured at much lower excitation power density in comparison with that used by Cai et al., also supports this attribution. The absence of this peak in the sample #*GT1* grown at T_S_ ≈ 450 °C can be associated with much worse structural perfection of #*GT1* layer compared to #*GT2*. The peak at 1.25 eV, which is the only one registered in the PL spectra of both samples (#*GT1* and #*GT2*), probably corresponds to the radiation of a gallium vacancy complex in the GaAs substrate [119]. This assumption was confirmed by observation of the same peak in the PL spectrum of the bare GaAs substrate without the GaTe layer. Nevertheless, this peak can also be related to the defect emission in GaTe, since the PL intensity at 1.25 eV in sample #*GT2* is by an order of magnitude higher.

The GaSe/GaTe system is perspective for the fabrication of QW structures due to a type I band alignment at GaSe/GaTe interface [120]. The GaSe/GaTe/GaSe heterostructure was grown on GaAs(001) substrate with a 200-nm-thick GaAs buffer layer. To provide the relatively flat growth surface for the GaTe deposition, the bottom GaSe layer was grown at relatively low T_S_ = 400 °C with Se/Ga(BEP) ~ 20 flux ratio. The average growth rate of GaSe was as low ~1.3 nm/min. GaTe layer was deposited at T_S_ = 450 °C, using the similar growth conditions as in sample #*GT1*. The growth temperature of the top GaSe was chosen also equal to T_S_ = 400 °C to avoid the Se diffusion into the GaTe layers. The TEM image of the GaSe/GaTe/GaSe heterostructure is presented in Figure 18b. It can be seen that the GaTe layer differs in contrast from GaSe and forms a rather uniform layer. The interfaces between 2D materials are rather sharp. However, no PL has been revealed from the GaSe/GaTe/GaSe QW heterostructure. We assume that the main reason for the absence of PL in GaTe QW is a low growth temperature of both GaTe QW and GaSe barriers.

## 4. Conclusions

GaSe, InSe, and GaTe layers as well as the relevant heterostructures were grown by MBE on the GaAs(001) substrates at T_S_ = 400–530 °C using a standard Ga, In, and Te effusion cells as well as a Se valve cracking cell with the cracking zone temperature T_Se_(cr) = 500 °C as molecular beam sources. The structures were studied in detail using SEM, TEM, X-ray diffraction, Raman, and PL spectroscopy techniques. The growth conditions corresponding to the stoichiometric ratio of Ga and Se adatoms on the growth surface as a function of growth temperature, as well as the correlation between the growth conditions and the properties of the GaSe layers were established. Powder X-ray diffraction has shown that the GaSe layers grown under strong Se-rich conditions contain inclusions of the secondary Ga_2_Se_3_ phase. A combination of SAED and Raman spectroscopy techniques confirmed that the GaSe layers with a relatively flat surface morphology grown at T_S_ = 400 °C correspond to the γ-GaSe polytype, and the ***c*** axis of the GaSe layer in this case is normal to the substrate/layer interface. Contrary to that, GaSe layers grown at high T_S_ = 500 °C on the GaAs(001) substrates predominantly possess an ε-GaSe polytype, and the TEM images indicate the preferential orientation of the ***c*** axis of ε-GaSe along the <111> directions of the GaAs substrate due to the chemical interaction between the substrate and the growing layer. Nevertheless, the formation of a particular GaSe-polytype is governed not only by the growth temperature, but rather by a combination of various factors. The RT near band-edge excitonic photoluminescence in the GaSe/GaAs(001) layers grown at T_S_ ~ 500 °C was demonstrated, while the intense emission band centered at ~1.7 eV dominates the PL spectra of the GaSe/GaAs(001) layers grown at low T_S_ ~ 400 °C temperature.

The MBE growth of InSe directly on the GaAs(001) was found to be a challenge due to formation of numerous In_x_Se_y_-type phases in the InSe film. In the low-temperature (T = 11 K) PL spectra of InSe/GaAs(001) layers, the highest-energy peak was detected at ~1.307 eV. The temperature dependence of the PL spectrum revealed that this peak consists of the two closely-spaced and poorly separated bands, which were attributed to the recombination of native acceptor and donor–acceptor pairs, respectively. Taking account of difficulties in InSe/GaAs(001) heteroepitaxy, we tried to realize the ZnSe/InSe/ZnSe quantum well (QW) heterostructures with the thickness of InSe layers varied within 2–7 TLs range. The TEM data reveal the island character of the InSe/ZnSe(001) nucleation; however, all QW structures demonstrated the strong intense broad PL band centered at ~2.27 eV, regardless of the QW width. This PL band was associated with radiative recombination centers of In in ZnSe because of the In diffusion into the underlying ZnSe layer. In addition, the strong diffusion of Ga into the top InSe layer was observed in InSe/GaSe/GaAs(001) structures at InSe growth temperature of T_S_ ~ 450 °C.

The near band-edge PL at T = 11 K was observed in GaTe layer grown at high T_S_ ~ 530 °C. The peak with energy of ~1.72 eV was associated with the emission of excitons bound at the acceptor in GaTe. However, no excitonic PL was observed in the GaSe/GaTe/GaSe QW heterostructure, probably due to low growth temperature of both GaTe QW and GaSe barriers, suggesting the thesis that growth temperature is the main factor affecting the optical quality of the III–VI layers. Nevertheless, the GaSe/GaTe material system looks as the most prospective for the fabrication of the optically active QW structures due to the ability to grow the relatively flat GaTe layers at high growth temperature T_S_ > 500 °C.

## Figures and Tables

**Figure 1 materials-13-03447-f001:**
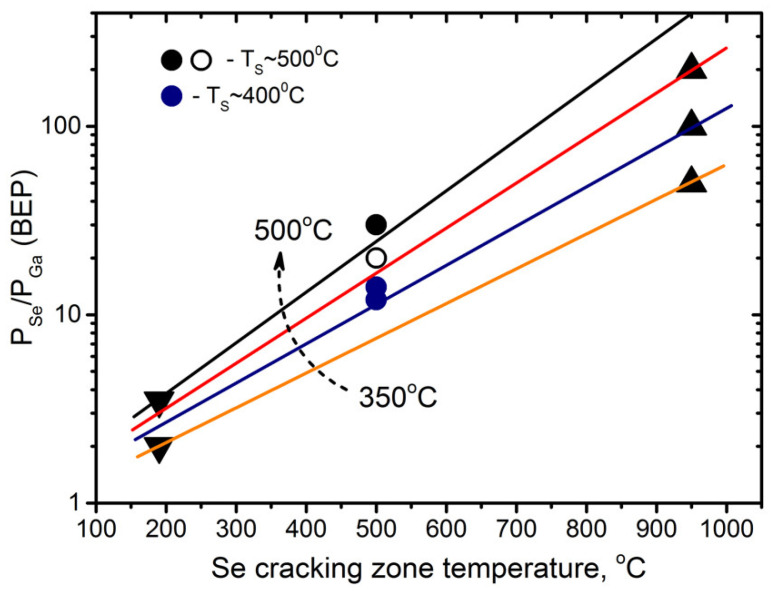
The calculated dependences of the P_Se_/P_Ga_ (BEP) ratio corresponding to stoichiometric conditions on the GaSe growth surface at different T_S_, marked in different colors (T_S_ = 350 °C, orange; T_S_ = 400 °C, blue; T_S_ = 450 °C, red; and T_S_ = 500 °C, black). The data of Lee et al. [30] are shown by filled triangles with the vertex pointing up (T_Se_(cr) = 950 °C). The data on P_Se_/P_Ga_ (BEP) flux ratios corresponding to nearly stoichiometric conditions for GaSe layers growth using standard Ga and Se cells are shown by filled triangles with the vertex pointing down. Our data on P_Se_/P_Ga_ (BEP) flux ratio for the GaSe layers grown at T_S_ = 400 and 500 °C are shown by blue and black circles, respectively. GaSe layers with and without Ga droplets on the growth surface are indicated by the hollow and filled circles.

**Figure 2 materials-13-03447-f002:**
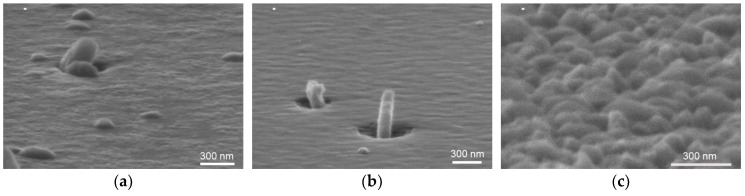
SEM images of the GaSe layers grown on the epi-ready GaAs(001) substrates at T_S_ ~ 400 °C. The Se/Ga (BEP) flux ratios and growth rates were the following: (**a**) Se/Ga (BEP) ~ 42, r_GaSe_ ≈ 1.13 nm/min, sample #*GS2*; (**b**) Se/Ga(BEP) ~ 34, r_GaSe_ ≈ 1.3 nm/min, #*GS1*; (**c**) Se/Ga (BEP) ~ 12–13, r_GaSe_ ≈ 5 nm/min, #*GS5*. The thicknesses of the structures ranged within 200–300 nm. In the case of using standard Ga and Se valve cracking cell with T_Se_(cr) = 500 °C as molecular beam sources, the stoichiometric Se/Ga ratio at T_S_ = 400 °C corresponds to the P_Se_/P_Ga_ (BEP) ~ 12.

**Figure 3 materials-13-03447-f003:**
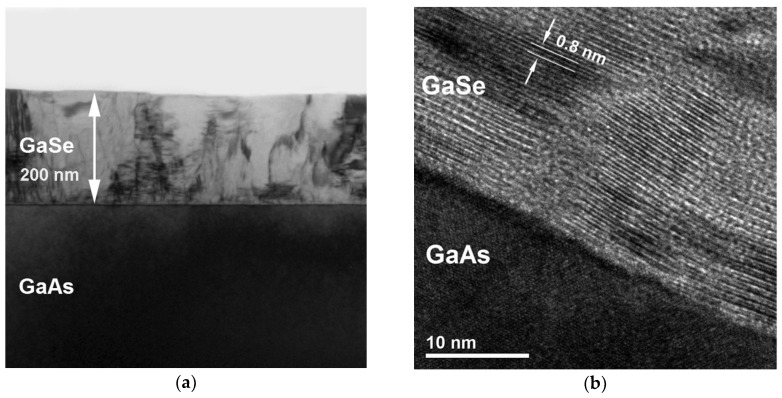
(**a**) Cross-section TEM image of the GaSe layer grown on the GaAs(001) substrate at T_S_ ~ 400 °C (Se/Ga(BEP) ~ 34) (sample #*GS1*); (**b**) high-resolution TEM (HRTEM) image of the GaSe/GaAs(001) heterointerface for the same layer.

**Figure 4 materials-13-03447-f004:**
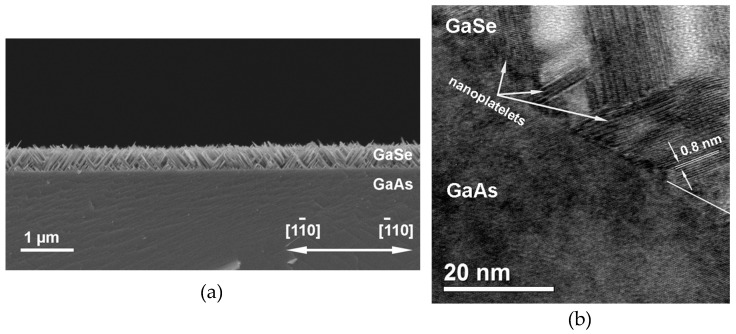
Cross-section SEM (**a**) and HRTEM (**b**) images of the GaSe layer grown on a GaAs(001) substrate at T_S_ ~ 500 °C (Se/Ga (BEP) ~ 25) (sample #*GS7*).

**Figure 5 materials-13-03447-f005:**
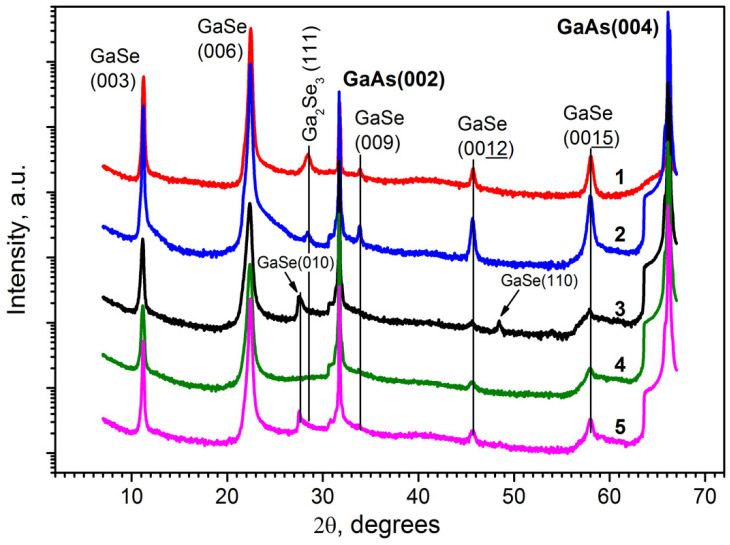
The X-ray powder diffraction patterns of the GaSe layers grown on GaAs(001) substrates. A weak additional peak in the angle range ~28.4°, along the 2θ axis on XRD patterns of the layers grown under strong Se-rich conditions at T_S_ = 400 °C (Curves 1 and 2), is attributed to the inclusions of a Ga_2_Se_3_ phase. Curves 1–5 correspond to samples #*GS2*, #*GS1*, #*GS5*, #*GS4*, and #*GS3*, respectively.

**Figure 6 materials-13-03447-f006:**
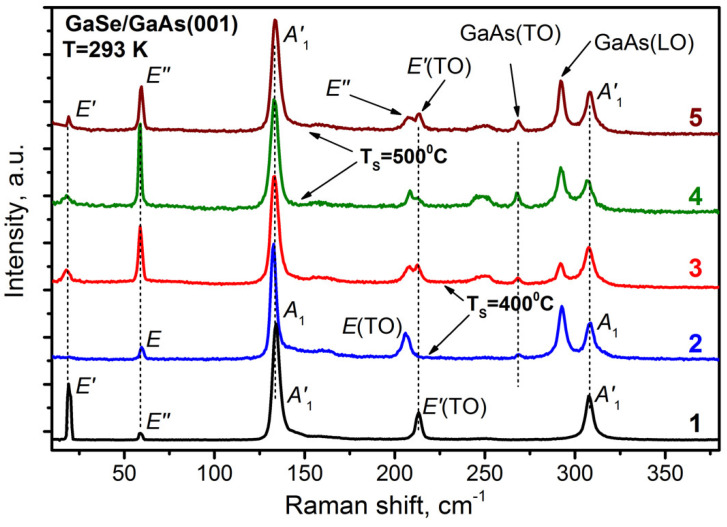
Raman spectrum of the bulk ε-GaSe layer produced by HQ Graphene (Curve 1). Raman spectra of the GaSe layers grown on GaAs(001) substrates at T_S_~400 °C (Curves 2 and 3, samples #*GS1* and #*GS8*, respectively), at T_S_ ~ 500 °C (Curve 5, sample #*GS7*), and using a two-stage growth mode (Curve 4, sample #*GS3*). All spectra are normalized to the intensity of the A_1_ phonon line (132 cm^−1^).

**Figure 7 materials-13-03447-f007:**
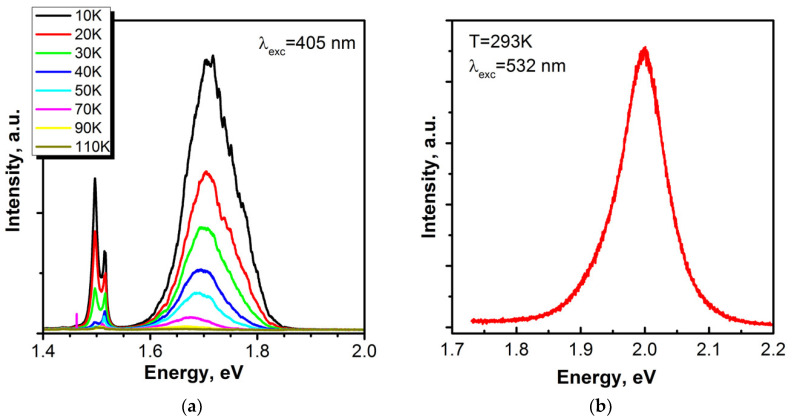
(**a**) μ-PL spectra of the GaSe layer grown at T_S_ = 400 °C as a function of temperature (sample #*GS1*). The laser power density on the sample was below 1 W/cm^2^. (**b**) PL spectrum of the GaSe layer grown at T_S_ = 500 °C (sample *#GS7*).

**Figure 8 materials-13-03447-f008:**
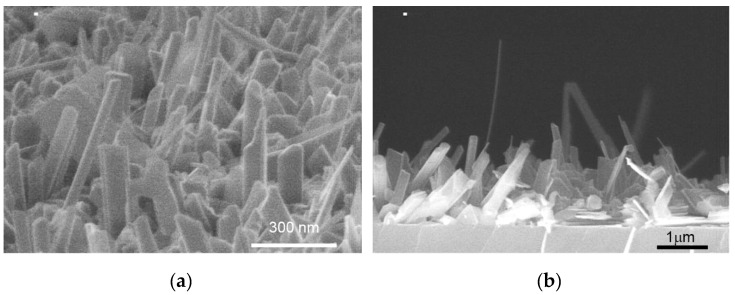
The plan-view (**a**) and cross-section (**b**) SEM images of the InSe/GaAs(001) film (sample #*IS1*) grown at T_S_ = 450 °C at nearly stoichiometric growth conditions.

**Figure 9 materials-13-03447-f009:**
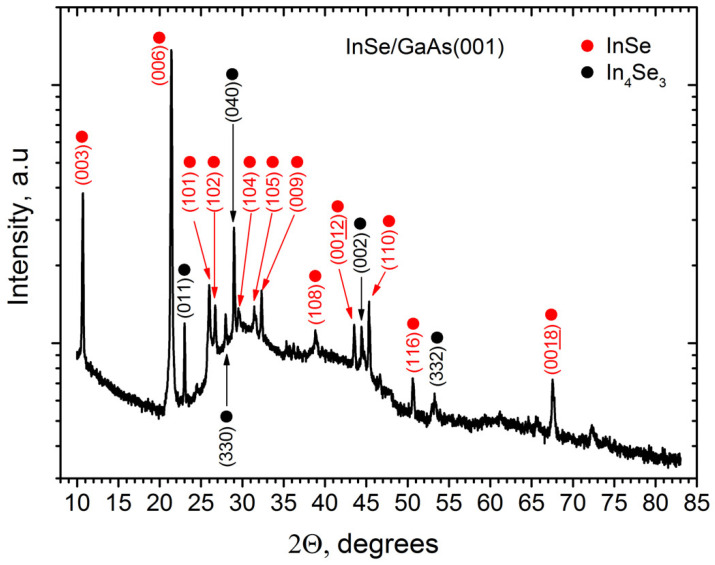
The X-ray powder diffraction pattern of the InSe/GaAs(001) layer (sample #*IS1*). Reflections corresponding to InSe are indicated by red circles, while reflections that can be attributed to the In_4_Se_3_ phase are marked by black circles. The InSe reflections are indexed for the γ-InSe rhombohedral lattice.

**Figure 10 materials-13-03447-f010:**
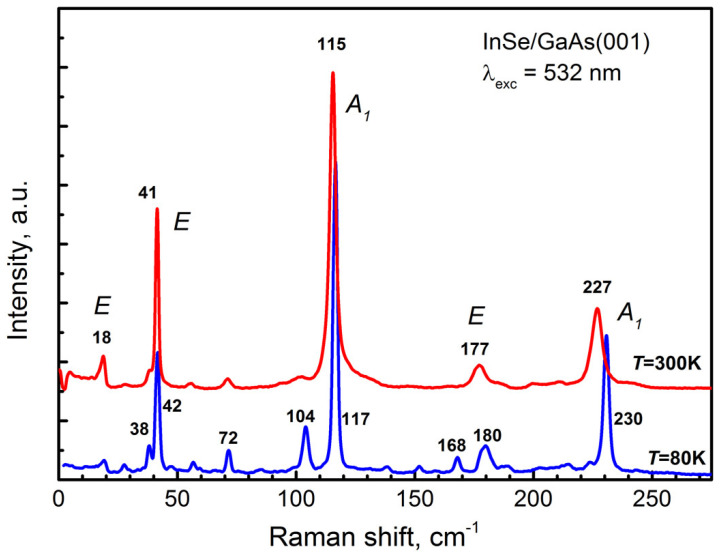
The Raman spectra of the InSe/GaAs(001) layer (sample #*IS1*) measured at 300 K (red curve) and at 80 K (blue curve). The symmetrycep of the phonon modes are given in accordance with [87].

**Figure 11 materials-13-03447-f011:**
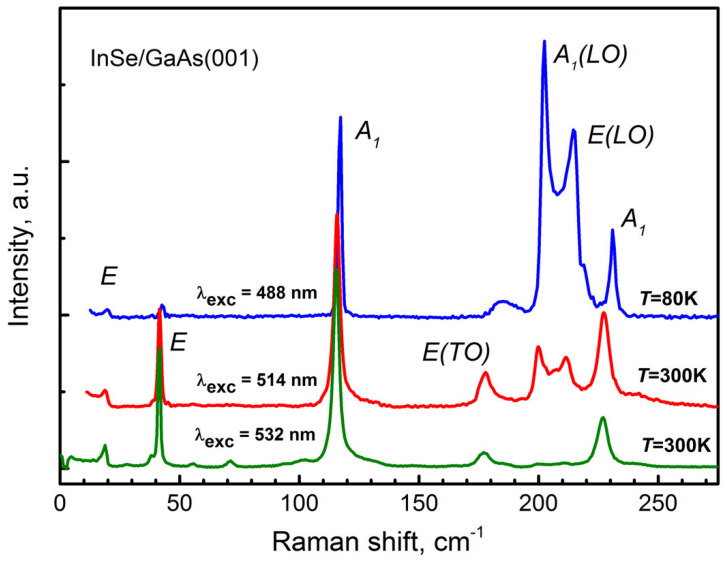
The near-resonance Raman spectra of the InSe/GaAs(001) layer (sample #*IS1*) measured at 300 K (λ_exc_ = 514.5 nm, red curve) and at 80 K (λ_exc_ = 488 nm, blue curve) along with the off-resonance Raman spectrum of the same layer measured at 300 K (λ_exc_ = 532 nm, green curve). The spectra are normalized to the intensity of the A_1_ (115 cm^−1^) phonon line.

**Figure 12 materials-13-03447-f012:**
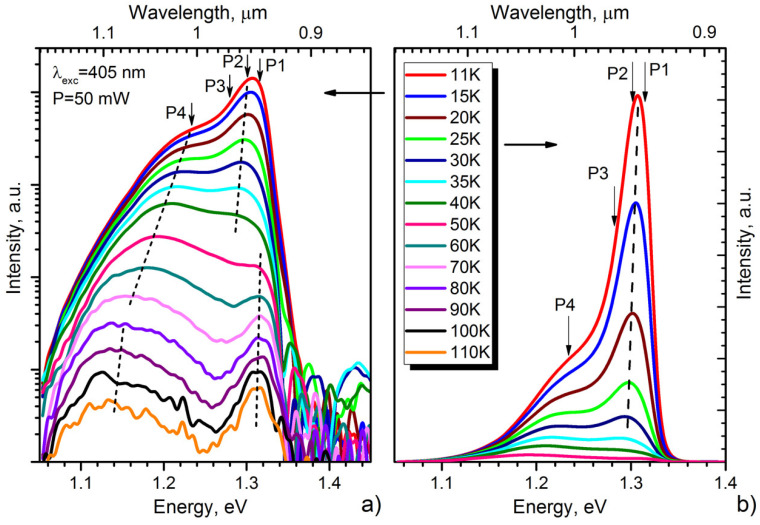
Photoluminescence spectra of the InSe/GaAs(001) layer (sample #*IS1*) at a low excitation power of 0.5 W/cm^2^ in logarithmic (**a**) and linear (**b**) scales measured at different temperatures. The colors of the curves correspond to the legend presented in (**b**). The *P1*, *P2*, *P3* and *P4* peaks correspond to those shown in Appendix A. The dashed lines are shown for eyes only and correspond to temperature shift of the respective peak maxima.

**Figure 13 materials-13-03447-f013:**
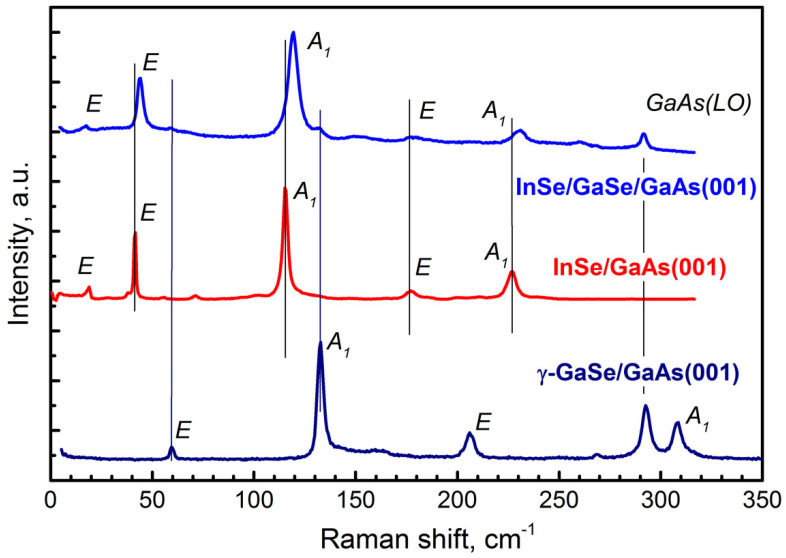
The Raman spectrum (upper curve) of the InSe/GaSe/GaAs(001) structure along with the Raman spectra of both InSe/GaAs(001) (sample #*IS1*, middle curve) and γ-GaSe/GaAs(001) (sample #*GS1*, lower curve) layers (T = 300 K). The spectra are normalized to the intensity of the A_1_ (115 cm^−1^) phonon line.

**Figure 14 materials-13-03447-f014:**
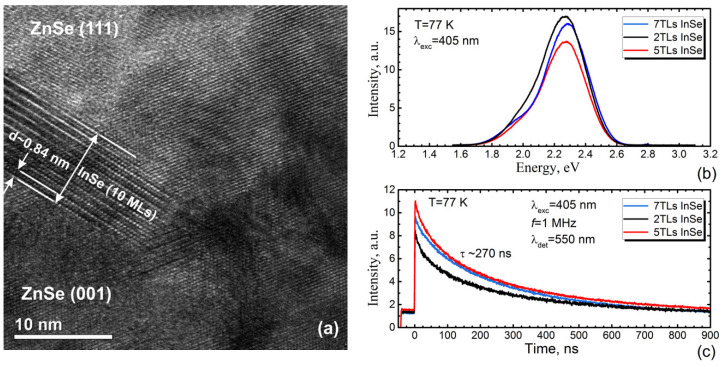
(**a**) Cross-section HRTEM image of the ZnSe/InSe/ZnSe QW structure with the InSe nominal thickness of ~7 TLs; (**b**) PL spectra of ZnSe/InSe/ZnSe QW structures with different InSe nominal thickness; (**c**) time-resolved measurements of the luminescence in ZnSe/InSe/ZnSe QW structures at T = 77 K. The samples were excited by 405 nm radiation of the CUBE laser (Coherent) with pulses of 45 ps duration at a repetition rate of 1 MHz.

**Figure 15 materials-13-03447-f015:**
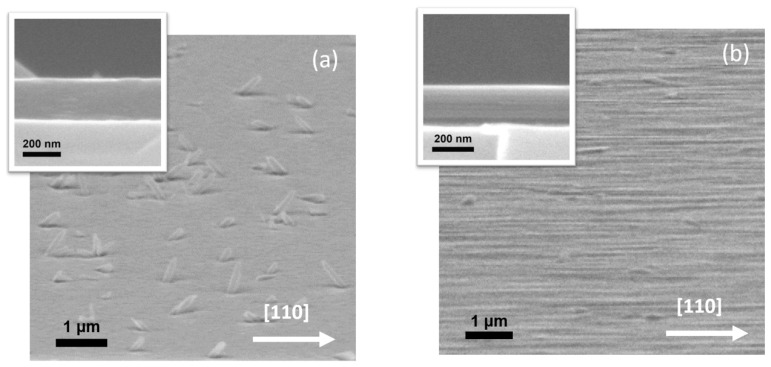
Plan-view SEM images of the GaTe layers grown on GaAs(001) substrates at different substrate temperature: (**a**) sample #*GT1*, T_S_ ≈ 450 °C; (**b**) sample #*GT2*, T_S_ ≈ 530 °C. The insets show the cross-section SEM images of corresponding structures.

**Figure 16 materials-13-03447-f016:**
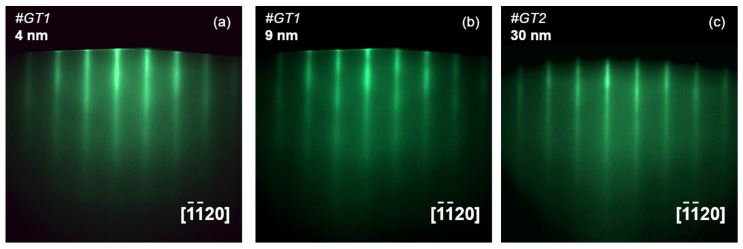
The evolution of the RHEED patterns along the $\left[\overset{¯}{1}\overset{¯}{1}20\right]$ direction in MBE of GaTe/GaAs(001) epilayers grown at low T_S_ ≈ 450 °C (sample #*GT1*): (**a**) the layer thickness of ~4 nm; (**b**) the layer thickness of ~ 9 nm; (**c**) the RHEED pattern of sample #*GT2* grown at T_S_ ≈ 530 °C (the layer thickness is ~30 nm).

**Figure 17 materials-13-03447-f017:**
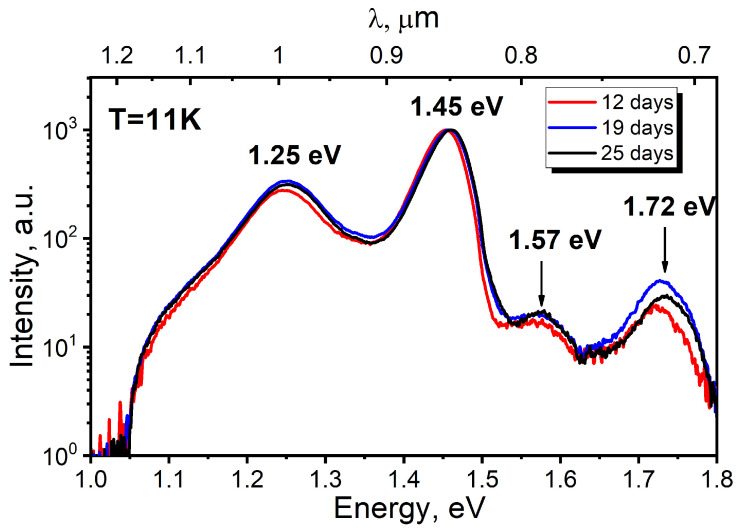
PL spectra of the GaTe/GaAs(001) layer #*GT2* measured after different storage time in vacuum (within three weeks). The PL intensity is normalized to the PL peak with energy of 1.45 eV.

**Figure 18 materials-13-03447-f018:**
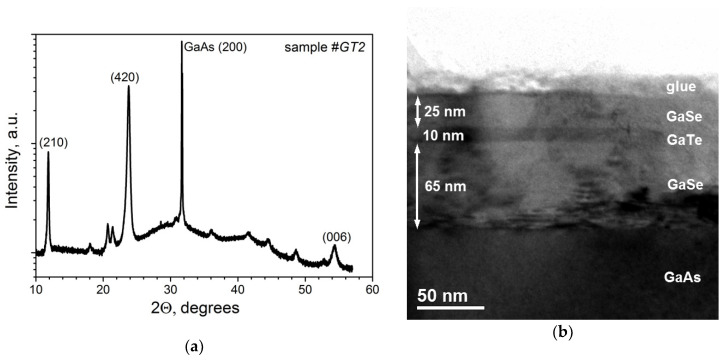
(**a**) The X-ray powder diffraction pattern of GaTe layer grown at T_S_ ≈ 530 °C (#*GT2*); (**b**) cross-section TEM image of the GaSe/GaTe/GaSe heterostructure.

**Table 1 materials-13-03447-t001:** The growth parameters of GaSe/GaAs(001) layers.

Layer	T_Ga_; Ga flux (BEP)	Se/Ga Flux Ratio ^1^, (BEP/Real)	*T*_S_, °C	Layer Thickness, nm (SEM)	Average Growth Rate ^2^, nm/min	Notes
#*GS1*	T_Ga_ = 910 °C; P_Ga_ = 4.7 × 10^−8^ Torr	Se/Ga (BEP) ~ 34; (Se/Ga ~ 3.9)	410	200	1.3	Se cracking zone temperature *T_Se_*(cr) = 450 °C
#*GS2*	T_Ga_ = 900 °C; P_Ga_ = 4.0 × 10^−8^ Torr	Se/Ga (BEP) ~ 42; (Se/Ga ~ 4.8)	400	270	1.13	Se cracking zone temperature *T_Se_*(cr) = 450 °C
#*GS3*	T_Ga_ = 940 °C (1st stage); T_Ga_ = 920 °C (2nd stage); P_Ga_ = 9.0 × 10^−8^ Torr (1st stage); P_Ga_ = 6.0 × 10^−8^ Torr (2nd stage)	Se/Ga (BEP) ~ 20/30; (Se/Ga ~ 1.6/1.2)	410—1st stage 500—2nd stage	360	~3	0.2-μm thick GaAs buffer layer; two-stage growth mode
#*GS4*	T_Ga_ = 940 °C; P_Ga_ = 9.0 × 10^−8^ Torr	Se/Ga (BEP) ~ 18–20; (Se/Ga ~ 0.9–0.95)	480	50	~1.4	Ga droplets on the surface with diameter ~200–300 nm
#*GS5*	T_Ga_ = 954 °C; P_Ga_ = 1.25 × 10^−7^ Torr	Se/Ga (BEP) ~ 12–13; (Se/Ga ~ 1)	410	300	5	0.2-μm thick GaAs buffer layer
#*GS6*	T_Ga_ = 910 °C; P_Ga_ = 5.5 × 10^−8^ Torr	Se/Ga (BEP) ~ 13–14; (Se/Ga ~ 1.1–1.2)	405	90	1.5	
#*GS7*	T_Ga_ = 910 °C; P_Ga_ = 5.5 × 10^−8^ Torr	Se/Ga (BEP)~25; (Se/Ga~1)	500	400	2.2	
#*GS8*	T_Ga_ = 890 °C; P_G a_= 3.8 × 10^−8^ Torr	Se/Ga (BEP)~21; (Se/Ga~1.6)	410	-	~1.0	rough growth surface

^1^ The calculated Se/Ga ratios of the impinging atoms on the GaAs surface are given taking into account the growth temperature (T_S_) and the cracking zone temperature of the valve Se source, influencing the Se sticking coefficient. ^2^ The average growth rate was calculated using the layer thickness determined from SEM measurements and deposition time.

**Table 2 materials-13-03447-t002:** The growth parameters of GaTe/GaAs (001) layers.

Layer	T_Ga_; Ga Flux (BEP)	Te/Ga Flux Ratio, (BEP)	*T*_S_, °C	Layer Thickness, nm (SEM)	Average Growth Rate^2^, nm/min	Notes
#*GT1*	900 °C; 4.0 × 10^−8^ Torr.	Te/Ga (BEP) ~ 11	450	230	1.28	0.2-μm-thick GaAs buffer layer
#*GT2*	920 °C; 6 × 10^−8^ Torr.	Te/Ga (BEP) ~ 15	530	200	1.82	0.2-μm-thick GaAs buffer layer

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
