# Peer review of "Molecular Beam Epitaxy of Layered Group III Metal Chalcogenides on GaAs(001) Substrates"

_materials, 2020, doi:10.3390/ma13163447_

Round 1

Reviewer 1 Report

The authors have reported the growth of layered metal (of group III) chalcogenides on GaAs(001) substrates. They have studied the grown films using the characterization techniques like – XRD, Raman & PL spectroscopy and TEM in order to understand the relation between film properties and the growth conditions. The reviewer is very pleased to see the details with which the authors have conducted this research. However, there are a few points that the reviewer would like to bring to the attention of the authors for further inputs or corrections –

In Figure 16 a), b) and c) - GT1 and GT2 are of different thicknesses. What is the reason for selecting different thicknesses? What is observed in the RHEED patterns if the thicknesses are identical for 450C and 530C films.

Please elaborate line 733-734 – “Following the Cai et al. [118], we can suppose that this peak can be associated with the emission centers localized at the domain edges.”

In Figure 17 --> Please include the Legend for the three colors used. It would be good to show the PL spectra of #GT1 as well as an overlap with #GT2.

In Figure 18b --> what is the label of the layer above 25nm GaSe.

How is the low growth temperature corelated to the absence of PL signal in GaSe/GaTe/GaSe heterostructure?

Line 764 – ‘adatams’ --> adatoms.

Author Response

Reviewer 1

Dear Reviewer,

We are grateful to you for the careful reading of our manuscript and valuable comments. In accordance with your criticism, we have made the following corrections to the text of the manuscript:

  1. In Figure 16 a), b) and c) - GT1 and GT2 are of different thicknesses. What is the reason for selecting different thicknesses? What is observed in the RHEED patterns if the thicknesses are identical for 450°C and 530°C films.

Indeed, in Figure 16 we show the RHEED data for samples #GT1 and #GT2 recorded at different thicknesses. The main reason is that the brightness of the RHEED pattern of sample #GT1 grown at low TS=450°C gradually decreased with increasing layer thickness (see also line 703 of the manuscript). The streaky RHEED pattern persists until the end of growth, but the intensity of the reflexes lowers down (see images below), indicating that the growth conditions are not optimal. This transformation of RHEED pattern of #GT1 sample is followed in Figures 16a and 16b of the Manuscript. In case of sample #GT2, we did not observe a decrease in the RHEED pattern intensity with time, so the RHEED pattern recorded at the thicker layer of ~30 nm has been shown.

  1. Please elaborate line 733-734 – “Following the Cai et al. [118], we can suppose that this peak can be associated with the emission centers localized at the domain edges.”

We rewrote the sentence.

“Cai et al. [118] argued that this peak is associated with the emission centers localized at the domain edges. The dominance of this emission band (1.45 eV) in our PL spectrum, measured at much lower excitation power density in comparison with that used by Cai et al., also supports this attribution.”

  1. In Figure 17 --> Please include the Legend for the three colors used. It would be good to show the PL spectra of #GT1 as well as an overlap with #GT2.

We have slightly changed Figure 17 by adding legend to the Figure according to the Reviewer recommendation.

Figure 17 (revised version).

The PL spectrum of sample #GT1 along with the spectrum of sample #GT2 has been added into the Supplementary Information as Figure S8.

Figure S8. PL spectra of the GaTe/GaAs(001) layers #GT1 and #GT2 measured at the same excitation power density of 0.5 W/cm2.

  1. In Figure 18b --> what is the label of the layer above 25nm GaSe.

There is no additional layer above GaSe in Figure 18b. It is an artifact of the TEM sample preparation, in particular, the adhesive (glue). It is seen, that this “layer” is not flat. The revised version of Figure 18b is presented and incorporated into the Manuscript.

Figure 18b (revised version).

  1. How is the low growth temperature correlated to the absence of PL signal in GaSe/GaTe/GaSe heterostructure?

For all 2D layers grown (GaSe, GaTe, InSe), we observed that the higher the growth temperature, the higher is the optical quality of the layers. In this work, we demonstrated the difference in PL spectra for GaTe layers grown at low (TS=450°C) and high (TS=530°C) growth temperatures. It was surprising for us that the flat continuous GaTe layers can be deposited on GaAs(001) substrates at such a high growth temperature, which is completely different from the case of GaSe/GaAs(001) epitaxy. In the GaSe/GaTe/GaSe QW heterostructure (Figure 18b), the GaTe layer was deposited at low TS=450°C. It was assumed that if we did not observe the PL signal from the GaTe thick layer grown at the same TS=450°C, then this is the likely reason of the absence of PL signal from a GaTe QW grown under the same conditions. However, for the QW structure with GaTe layer grown at high TS=530°C, the situation may be different, although the growth of such a structure is a challenge due to the rough platelet morphology of GaSe layer at this temperature.

  1. Line 764 – ‘adatams’ --> adatoms.

We have made this correction.

We have also found a couple of misprints in the Manuscript and made the minor corrections in the text:

Line 85. The GaSe lattice parameter should be c=23.92Å.

In caption to Figure 12, 6 W/cm2 should be changed by 0.5 W/cm2.

We hope that the revised version of the manuscript satisfies now the requirements necessary for its publication.

Sincerely yours,

S.V. Sorokin

Reviewer 2 Report

The manuscript deals with an interesting topic, which is to analyze the parameters effect GaSe, InSe, GaTe and their heterostructure MBE growth. The authors did a great job when designing the experiment and the results are interesting and well explained. The sections of introduction and materials properties present good literature review, it is clear, and helps justifying the objective of the study, which is desired. The conclusions agree well with the results and summarize well the main findings of the study.

However, I realized some figures (e,g. 3(b), 4, 6) are from the authors’ other published article (ref 74 in this manuscript). It is not appropriate to directly use these figures without any update or figure copyright.

There are also some minor recommendations and comments as followed:

  1. Without the use of GaAs, what about the critical thickness of GaSe and InSe?
  2. In figure 1, the temperature of each line is not explicitly indicated.
  3. In figure 5, what about the peak at 48in curve 3?
  4. In line 393-394 and figure 6, does curve-2 stand for #GS2 or #GS3?
  5. As the statement in section 3.2, the InSe/GaAs(001) growth is difficult at ~450C and higher temperature. Do the author try some lower temperature like ~400C
  6. In figure 9, there seems to be a large amorphous background from XRD. Is it possible some amorphous InSe exist during the growth?
  7. The legend of figure 10, 11, 12a and 17 are missing.
  8. The location of P1, P2, P3 and P4 and their shift (dash lines?) in figure 12.
  9. In Figure 17 and 18, for comparison of #GT1 and #GT2, the PL and XRD data of #GT1 are missing.
